# Unveiling Transformer Perception by Exploring Input Manifolds

**Alessandro Benfenati**[1a]  **Alfio Ferrara**[1b]  **Alessio Marta**[1c]  **Davide Riva**[2d]  **Elisabetta Rocchetti**[1b*]

[a]Department of Environmental Science and Policy, [b]Department of Computer Science,
[c]Department of Mathematics, [d]Department of Control and Computer Engineering
[1]Università degli Studi di Milano, [2]Politecnico di Torino
{alessandro.benfenati, alfio.ferrara}@unimi.it
{alessio.marta, elisabetta.rocchetti}@unimi.it
{davide.riva}@polito.it
[*] Corresponding author

## Abstract

This paper introduces a general method for the exploration of equivalence classes in the input space of Transformer models. The proposed approach is based on sound mathematical theory which describes the internal layers of a Transformer architecture as sequential deformations of the input manifold. Using eigendecomposition of the pullback of the distance metric defined on the output space through the Jacobian of the model, we are able to reconstruct equivalence classes in the input space and navigate across them. Our method enables two complementary exploration procedures: the first retrieves input instances that produce the same class probability distribution as the original instance—thus identifying elements within the same equivalence class—while the second discovers instances that yield a different class probability distribution, effectively navigating toward distinct equivalence classes. Finally, we demonstrate how the retrieved instances can be meaningfully interpreted by projecting their embeddings back into a human-readable format. **Disclaimer**: This paper includes examples of sensitive and very offensive language solely to illustrate the behavior of LLMs exploring the input space.

## 1   Introduction

In the literature, the investigation of the input space of Transformers relies on perturbations of input data using heuristic or gradient-based criteria [24, 17, 14], or on the analysis of specific properties of the embedding space [6] via the production of optimal robust explanations and counterfactuals. In this paper, we propose a method for exploring the input space of Transformer models by identifying *equivalence classes* with respect to their predictions. Our approach is based on sound mathematical theory which describes the internal layers of a Transformer architecture as sequential deformations of the input manifold. Using eigendecomposition of the pullback of the distance metric defined on the output space through the Jacobian of the model, we are able to reconstruct equivalence classes in the input space and navigate in and across them. The equivalence class consists of the counterimage of a particular probability distribution: this means that the elements of an equivalence class, once fed to the Transformer, will provide different realizations from the same probability distribution. Thanks to our approach, we provide two different methods for exploring the embedding space: the Singular Metric Equivalence Class (SiMEC) and the Singular Metric Exploration (SiMExp) algorithms. The first allows for the identification of inputs within the same equivalence class. This means that, given two data inputs $x, x'$ identified through the exploration process as belonging to the same equivalence class and a class label $C$, our method guarantees that the Transformers will assign the same probability, modulo numerical approximations, to that label for both inputs: $p(C|x) \approx p(C|x')$.

39th Conference on Neural Information Processing Systems (NeurIPS 2025).

The second method, instead, allows for the exploration of the embedding space starting from an element within an equivalence class and moving towards a different equivalence class. This means that, given two inputs, namely $x, x'$, identified in this way and a class label $C$, we guarantee that the Transformer will assign different probabilities to that label for the two inputs, *i.e.*, $p(C|x) \neq p(C|x')$. Our experimental data show that this different probability assignment can lead to a change in the most probable class in the Transformers prediction.

In Section 2, we summarise the mathematical foundations of our approach. In Section 3, we present our method for the exploration of equivalence classes in the input of the Transformer models. In Section 4, we empirically investigate the effectiveness and applicability on Transformer models on textual and visual data. In Section 5, we discuss the relevant literature on embedding space exploration. Finally, in Section 6, we give our concluding remarks[1].

## 2  Preliminaries

We provide in this Section the theoretical foundation of the proposed approach, namely the Geometric Deep Learning framework based on Riemannian Geometry [1, 2].

A neural network is considered as a sequence of maps, the layers of the network, between manifolds, and the latter are the spaces where the input and the outputs of the layers belong to.

**Definition 1** (Neural Network). *A neural network is a sequence of $\mathcal{C}^1$ maps $\Lambda_i$ between manifolds of the form:*

$$M_0 \xrightarrow{\Lambda_1} M_1 \xrightarrow{\Lambda_2} M_2 \xrightarrow{\Lambda_3} \cdots \xrightarrow{\Lambda_{n-1}} M_{n-1} \xrightarrow{\Lambda_n} M_n \tag{1}$$

*We call $M_0$ the input manifold and $M_n$ the output manifold. All the other manifolds of the sequence are called representation manifolds. The maps $\Lambda_i$ are the layers of the neural network. We denote by $\mathcal{N}_{(i)} = \Lambda_n \circ \cdots \circ \Lambda_i : M_i \to M_n$ the mapping from the $i$-th representation layer to the output layer.*

As an example, consider a shallow network with just one layer, the composition of a linear operator $A \cdot + b$ with a sigmoid function $\sigma$, where $A \in \mathbb{R}^{m \times n}$ and $b \in \mathbb{R}^m$: then, the input manifold $M_0$ and the output manifold $M_1$ shall be $\mathbb{R}^n$ and $\mathbb{R}^m$, respectively, and the map $\Lambda_1(\cdot) = \sigma(A \cdot + b)$. We generalize this observation into the following definition.

**Definition 2** (Smooth layer). *A map $\Lambda_i : M_{i-1} \to M_i$ is called a smooth layer if it is the restriction to $M_{i-1}$ of a function $\overline{\Lambda}^{(i)}(x) : \mathbb{R}^{d_{i-1}} \to \mathbb{R}^{d_i}$ of the form $\overline{\Lambda}^{(i)}_\alpha(x) = F^{(i)}_\alpha \left( \sum_\beta A^{(i)}_{\alpha\beta} x_\beta + b^{(i)}_\alpha \right)$, for $i = 1, \cdots, n$, $x \in \mathbb{R}^{d_i}$, $b^{(i)} \in \mathbb{R}^{d_i}$ and $A^{(i)} \in \mathbb{R}^{d_i \times d_{i-1}}$, with $F^{(i)} : \mathbb{R}^{d_i} \to \mathbb{R}^{d_i}$ a diffeomorphism.*

**Remark 1.** *Transformers implicitly apply for this framework, since their modules are smooth functions, such as fully connected layers, GeLU and sigmoid activations, thus including also attention layers.*

Our aim is to transport the geometric information on the data lying in the output manifold to the input manifold: this allows us to obtain insight on how the network "sees" the input space, how it manipulates it for reaching its final conclusion. For fulfilling this objective, we need several tools from differential geometry. The first key ingredient is the notion of singular Riemannian metric, which has the intuitive meaning of a degenerate scalar product which changes point to point – the starting point for defining a non-Euclidean pseudodistance between points of a manifold.

**Definition 3** (Singular Riemannian metric). *Let $M = \mathbb{R}^n$ or an open subset of $\mathbb{R}^n$. A singular Riemannian metric $g$ over $M$ is a map $g : M \to Bil(\mathbb{R}^n \times \mathbb{R}^n)$ that associates to each point $p$ a positive semidefinite symmetric bilinear form $g_p : \mathbb{R}^n \times \mathbb{R}^n \to \mathbb{R}$ in a smooth way.*

Without loss of generality, we can assume the following hypotheses on the sequence (1): *i)* The manifolds $M_i$ are open and path-connected sets of dimension $\dim M_i = d_i$. *ii)* The maps $\Lambda_i$ are $\mathcal{C}^1$ submersions. *iii)* $\Lambda_i(M_{i-1}) = M_i$ for every $i = 1, \cdots, n$. *iv)* The manifold $M_n$ is equipped with the structure of Riemannian manifold, with metric $g^n$. Definition 3 naturally leads to the definition of the pseudolength and of energy of a curve.

---

[1]The code to reproduce our experiments can be found here:https://github.com/alessiomarta/transformers_equivalence_classes.

**Definition 4** (Pseudolength and energy of a curve). *Let $\gamma : [a, b] \to \mathbb{R}^n$ a curve defined on the interval $[a, b] \subset \mathbb{R}$ and $\|v\|_p = \sqrt{g_p(v, v)}$ the pseudo–norm induced by the pseudo–metric $g_p$ at point $p$. Then the pseudolength of $\gamma$ and its energy are defined as*

$$Pl(\gamma) = \int_a^b \|\dot{\gamma}(s)\|_{\gamma(s)} ds = \int_a^b \sqrt{g_{\gamma(s)}(\dot{\gamma}(s), \dot{\gamma}(s))} ds, \qquad E(\gamma) = \int_a^b \|\dot{\gamma}(s)\|_{\gamma(s)}^2 ds \quad (2)$$

The notion of pseudolength leads naturally to define the distance between two points.

**Definition 5** (Pseudodistance). *Let $x, y \in M = \mathbb{R}^n$. The pseudodistance between $x$ and $y$ is then*

$$Pd(x, y) = \inf\{Pl(\gamma) \mid \gamma : [0, 1] \to M, \ \gamma \in \mathcal{C}^1([0, 1]), \ \gamma(0) = x, \ \gamma(1) = y\}. \qquad (3)$$

One can observe that endowing the space $\mathbb{R}^n$ with a singular Riemannian metric leads to have non trivial curves whose length is zero. A straightforward consequence is that there are distinct points whose pseudodistance is therefore zero: a natural equivalence relation arises, *i.e.* $x \sim y \Leftrightarrow Pd(x, y) = 0$, obtaining thus a metric space $(\mathbb{R}^n / \sim, Pd)$.

The second crucial tool is the notion of pullback of a function. Intuitively, given a map $f : M \to N$ between two manifolds, the pullback operation allows to transfer the geometric information of the output $N$ onto the input $M$ by means of the Jacobian of $f$. More specifically, let $f$ be a function from $\mathbb{R}^p$ to $\mathbb{R}^q$, and fix the coordinate systems $x = (x_1, \ldots, x_p)$ and $y = (y_1, \ldots, y_q)$ on $\mathbb{R}^p$ and on $\mathbb{R}^q$, respectively. Moreover, we endow $\mathbb{R}^q$ with the standard Euclidean metric $g$, whose associated matrix is the identity. The space $\mathbb{R}^p$ can then be equipped with the pullback metric $f^*g$ whose representation matrix reads as:

$$(f^*g)_{ij} = \sum_{h,k=1}^q \left(\frac{\partial f_h}{\partial x_i}\right) g_{hk} \left(\frac{\partial f_k}{\partial x_j}\right). \qquad (4)$$

The sequence (1) shows that a neural network can be considered simply as a function, a composition of maps: hence, taking $f = \Lambda_n \circ \Lambda_{n-1} \circ \cdots \circ \Lambda_1$ and supposing that $M_0 = \mathbb{R}^p, M_n = \mathbb{R}^q$, the generalization of (4) applied to (1) provides with the pullback of a generic neural network.

Hereafter, we consider in (1) the case $M_n = \mathbb{R}^q$, equipped with the trivial metric $g^{(n)} = I_q$, *i.e.*, the identity. Each manifold $M_i$ of the sequence (1) is equipped with a Riemannian singular metric, denoted with $g^i$, obtained via the pullback of $\mathcal{N}_{(i)}$. The pseudolength of a curve $\gamma$ on the $i$-th manifold, namely $Pl_i(\gamma)$, is computed via the relative metric $g^i$ via (2).

## 2.1 General results

We depict hereafter the theoretical bases of our approach. We denote with $\mathcal{N}_i$ the submap $\Lambda_i \circ \cdots \circ \Lambda_n : M_i \to M_n$, and with $\mathcal{N} \equiv \mathcal{N}_0$ the map describing the action of the complete network. The starting point is to consider the pair $(M_i, Pd_i)$: this is a pseudometric space, which can be turned into a full-fledged metric space $M_i / \sim_i$ by the metric identification $x \sim_i y \Leftrightarrow Pd_i(x, y) = 0$. The first result states that the length of a curve on the $i$-th manifold is preserved among the mapping on the subsequent manifolds.

**Proposition 1.** *Let $\gamma : [0, 1] \to M_i$ be a piecewise $\mathcal{C}^1$ curve. Let $j \in \{i, i+1, \cdots, n\}$ and consider the curve $\gamma_j = \Lambda_j \circ \cdots \circ \Lambda_i \circ \gamma$ on $M_j$. Then $Pl_i(\gamma) = Pl_j(\gamma_j)$.*

In particular this is true when $k = n$, *i.e.*, the length of a curve is preserved in the last manifold. This result leads naturally to claim that if two points are in the same class of equivalence, then they are mapped into the same point under the action of the neural network.

**Proposition 2.** *If two points $p, q \in M_i$ are in the same class of equivalence, then $\mathcal{N}_i(p) = \mathcal{N}_i(q)$.*

The next step is to prove that the sets $M_i / \sim_i$ are actually smooth manifolds: to this aim, we introduce another equivalence relation: $x \sim_{\mathcal{N}_i} y$ if and only if there exists a piecewise $\gamma : [0, 1] \to M_i$ such that $\gamma(0) = x, \gamma(1) = y$ and $\mathcal{N}_i \circ \gamma(s) = \mathcal{N}_i(x) \ \forall s \in [0, 1]$. The introduction of this equivalence relation allows us to easily state the following proposition.

**Proposition 3.** *Let $x, y \in M_i$, then $x \sim_i y$ if and only if $x \sim_{\mathcal{N}_i} y$.*

The following corollary contains the natural consequences of the previous result; the second point of the claim below is the counterpart of Proposition 2.

**Corollary 1.** *Under the hypothesis of Proposition 3, one has that $M_i/\sim_i = M_i/\sim_{\mathcal{N}_i}$. Moreover, if two points $p, q \in M_i$ are connected by a $\mathcal{C}^1$ curve $\gamma : [0,1] \to M_i$ satisfying $\mathcal{N}_i(p) = \mathcal{N}_i \circ \gamma(s)$ for every $s \in [0,1]$, then they lie in the same class of equivalence.*

One is then able to prove that the set $M_i/\sim_i$ is a smooth manifold:

**Proposition 4.** $\dfrac{M_i}{\sim_i}$ *is a smooth manifold of dimension $dim(\mathcal{N}(M_0))$.*

This last achievement provides practical insights about the projection $\pi_i$ on the quotient space, that consists in the building block of the algorithms used for recovering and exploring the foliation in equivalence classes of a neural network.

**Proposition 5.** $\pi_i : M_i \to M_i/\sim_i$ *is a smooth fiber bundle, with $Ker(d\pi_i) = \mathcal{V}M_i$, which is therefore an integrable distribution. $\mathcal{V}M_i$ is the vertical bundle of $M_i$. Every class of equivalence $[p]$ is a path-connected submanifold of $M_i$ and coincide with the fiber of the bundle over the point $p \in M_i$.*

In [3] it is shown that these results keep to hold true in the case of convolutional layers and residual connections.

## 3 Methodology

The results depicted in Section 2.1 provide powerful tools for investigating how a neural network sees the input space starting from a point $x$. In particular we point out the following remarks: *i)* If two points $x, y$ belonging to the input manifold $M_0$ are such that $x \sim_0 y$, then $\mathcal{N}(x) = \mathcal{N}(y)$; *ii)* given a point $z \in M_n$, the counterimage $\mathcal{N}^{-1}(z)$ is a smooth manifold, whose connected components are classes of equivalence in $M_0$ with respect to $\sim_0$, then a necessary condition for two points $x, y \in M_0$ to be in the same class of equivalence is that $\mathcal{N}(x) = \mathcal{N}(y)$; *iii)* any class of equivalence $[x]$, $x \in M_0$, is a maximal integral submanifold of $\mathcal{V}M_0$. The above observations directly provide with a strategy to build up the equivalence class of an input point $x \in M_0$. Proposition 5 tells us that $\mathcal{V}M_0$ is an integrable distribution, with dimension equal to the dimension of the kernel of $g^0$: we can hence find $dim(Ker(g^0))$ vector fields which are a base for the tangent space of $M_0$. This means that we can compute the eigenvalue decomposition of $g_x^0$ and consider the $L$ linearly independent eigenvectors, namely $\{v_l\}_{l=1,\dots,L}$, associated to the null eigenvalue: these eigenvectors depend *smoothly* on the point, a fact that is not trivial when the matrix associated to the metric depends on several parameters [15]. We can build then all the null curves by randomly selecting one eigenvector $\tilde{v} \in \{v_l\}$ and then reconstruct the curve along the direction $\tilde{v}$ from the starting point $x$. From a practical point of view, one is led to solve the Cauchy problem with first order differential equation $\dot{\gamma} = \tilde{v}$ and initial condition $\gamma(0) = x$.

| **Algorithm 1** The Singular Metric Equivalence Class (SiMEC) algorithm. |
| --- |
| 1: Set the network $\mathcal{N}$; choose the number of iterations $K$. Choose the input $x^{(0)}$. |
| 2: **for** $k = 0, 1, \dots, K-1$ **do** |
| 3:      Compute $g_{\mathcal{N}(x^{(k)})}^n$ |
| 4:      Compute the pullback metric $g_{x^{(k)}}^0$ |
| 5:      Diagonalize $g_{x^{(k)}}^0$ and find the eigenvectors $\{v_l\}_{l \in L_0}$ associated to the zero eigenvalue $\lambda_0$ |
| 6:      Randomly select $\tilde{v} \in \{v_l\}_{l \in L_0}$ |
| 7:      $\delta^{(k)} = \eta \sqrt{\min_{j:\lambda_j \neq 0} |\lambda_j| / \max_j |\lambda_j|}$ |
| 8:      $x^{(k+1)} \leftarrow x^{(k)} + \delta^{(k)} \tilde{v}$ |
| 9: **end for** |
| 10: Project $x^{(k+1)}$ to the feasible region $\mathcal{X}$ |

| **Algorithm 2** The Singular Metric Exploration (SiMExp) algorithm. |
| --- |
| 1: Set the network $\mathcal{N}$; choose the number of iterations $K$. Choose the input $x^{(0)}$. |
| 2: **for** $k = 0, 1, \dots, K-1$ **do** |
| 3:      Compute $g_{\mathcal{N}(x^{(k)})}^n$ |
| 4:      Compute the pullback metric $g_{x^{(k)}}^0$ |
| 5:      Diagonalize $g_{x^{(k)}}^0$ and find the eigenvectors $\{w_l\}_{l \in L_+}$ associated to eigenvalues $\lambda_l \neq 0$ |
| 6:      Randomly select $\tilde{w} \in \{w_l\}_{l \in L_+}$ |
| 7:      $\delta^{(k)} = \eta \sqrt{\min_{j:\lambda_j \neq 0} |\lambda_j| / \max_j |\lambda_j|}$ |
| 8:      $x^{(k+1)} \leftarrow x^{(k)} + \delta^{(k)} \tilde{w}$ |
| 9: **end for** |
| 10: Project $x^{(k+1)}$ to the feasible region $\mathcal{X}$ |

## 3.1 Input Space Exploration

This entire procedure is coded in the Singular Metric Equivalence Class (SiMEC) and Singular Metric Exploration (SiMExp) algorithms, whose general schemes are depicted in Algorithms 1 and 2. SiMEC reconstructs the class of equivalence of the input via exploration of the input space by randomly selecting one of the eigenvectors related to the zero eigenvalue. On the opposite, in SiMExp, in order to move from a class of equivalence to another we consider the eigenvectors relative to the nonzero eigenvalues. This requires the slight difference in lines 5-6 between Algorithm 1 and Algorithm 2.

There are some remarks to point out. From a numerical point of view, the diagonalization of the pullback may lead to have even negative eigenvalues: hence one may use the notion of energy of a curve, related to the pseudolength. If the values $\delta^{(k)}$ are too small more iterations are needed to move away from the starting point sensibly. Therefore there is a trade-off between the reliability of the solution and the exploration pace. Relying on the theory of dynamical systems, we can in practice estimate $\delta^{(k)}$ at each iteration with the inverse of the square root of the condition number $\Gamma = \max_j |\lambda_j| / \min_{j:\lambda_j \neq 0} |\lambda_j|$ of the pullback metric $g^0_{x^{(k)}}$, as in a locally-linearized dynamical system. This is our default choice for both algorithms. We multiply the default value $\delta$ by a multiplier $\eta$ in order to explore the sensitivity of Algorithm 1 and Algorithm 2 to variations of step length, expecting Algorithm 1 to be more sensitive compared to Algorithm 2. Indeed, to build points in the same equivalence class Algorithm 1 needs to follow a null curve closely with as little approximation as possible. In contrast Algorithm 2, whose goal is to change the equivalence class, does not have the same problem and larger $\delta$ are allowed.

In the final step of each iteration of both algorithms, the embeddings $x^{(0)} \ldots x^{(K)}$ need to be constrained to a feasible region $\mathcal{X}$. This region is defined by the distribution of embeddings derived from the embedding layer, which is bounded by definition. Specifically its upper bound has components $UB_i = \sum_{j:E_{ij} >= 0} E_{ij} \max_l(x_l) + \sum_{j:E_{ij} < 0} E_{ij} \min_l(x_l) + \max_s(q(s)_i)$, where $E$ is the embedding layer weight matrix, $(x_l)$ represent input features, bounded 0 and 1 in both the visual and textual case, and $q(s)$ is the positional encoding vector at position $s$ in the sequence of patches/tokens. The lower bound $LB$ is obtained by switching max with min and vice versa. We acknowledge that these bounds present a non-null margin from the actual embedding distribution domain, however we find them to be a suitable estimate for the practical purpose of interpretation of input space exploration results, presented in next section. Notwithstanding numerical approximation errors, the outputs of SiMEC algorithm at each iteration $k$ are predicted by $\mathcal{N}$ to yield the same probability distribution $|p(\cdot|x^{(0)}) - p(\cdot|x^{(k)})| < \varepsilon$, $0 < \varepsilon \ll 1$, i.e. the original input probability, by construction. On the contrary, SiMExp algorithm induces a non-null probability change in the output space, which might possibly lead to a *class ranking change*, i.e. the situation where class A is given higher probability than class B at iteration $k - 1$ but it is surpassed by class B at iteration $k$, and eventually a *prediction change*, i.e. the situation in which $argmax(\mathcal{N}(x^{(k)})) = y' \neq argmax(\mathcal{N}(x^{(0)}))$ for $k$ in a non-degenerate neighborhood of a change-point iteration $\overline{k}$.

As for the computational complexity of the two algorithms, the most demanding step is the computation of the eigenvalues and eigenvectors, which is $O(n^3)$, with $n$ the dimension of the square matrix $g^0_{x^{(k)}}$ [20]. Since all the other operations are either $O(n)$ or $O(n^2)$, we conclude that the complexity of both Algorithms 1 and 2 is $O(n^3)$.

## 3.2 Interpretation

Algorithms 1 and 2 allow for the exploration of the equivalence classes in the input space of a Transformer model. However, the points explored by these algorithms may not be directly interpretable by a human perspective. For instance, an image or a piece of text may need to be decoded to be "readable" by a human observer. Here we present an interpretation method for Transformers based on input space exploration, which is then demonstrated on two Vision Transformer (ViT) models trained for image classification [8], and two BERT models, one trained for masked language modeling (MLM) [7] and the other fine-tuned for text classification [18].

Using SiMEC and SiMExp to explore the embedding space reveals how Transformer models perceive equivalence among different data points. Specifically, these methodologies facilitate the sequential acquisition of embedding matrices $x^{(0)} \ldots x^{(K)}$, as detailed in Algorithms 1 and 2. A key feature

of the SiMEC/SiMExp approach is its ability to selectively update specific tokens (for text inputs) or patches (for image inputs) during each iteration. This selective update allows to explore targeted modifications that prompt the model to either categorize different inputs as the same class or recognize them as distinct. Unlike other approaches [24, 12] where perturbations are predetermined, this method lets the model itself guide us to understand which data points belong to specific equivalence classes.

To interpret embeddings produced by the exploration process, they must be mapped back into a human-understandable form, such as text or images. The interpretation of an embedding vector depends on the operations performed by the Transformer's embedding module $\mathcal{E}_T$. If $\mathcal{E}_T$ consists only of invertible operations, it is feasible to construct a layer that performs the inverse operation relative to $\mathcal{E}_T$. The output can then be visualized and directly interpreted by humans, allowing for a comparison with the original input to discern how differences in embeddings reflect differences in their representations (e.g., text, images). If the operations in $\mathcal{E}_T$ are non-invertible, a trained decoder is required to reconstruct an interpretable output from each embedding matrix $x^{(1)}, \cdots, x^{(K)}$. Such operation injects some unforeseen noise into the interpretation results, which is investigated in the experimental setting. In practice, we construct the ViT models such that the embedding layer is invertible, whereas for BERT models it is feasible to exploit layers that are specialized for the MLM task to map input embeddings back to tokens. This approach is effective whether the BERT model in question is specifically designed for MLM or for sentence classification. In the case of sentence classification models, it is necessary to select a corresponding MLM BERT model that shares the same internal architecture, including the number of layers and embedding size.

Algorithm 3 depicts the process of interpreting SiMEC/SiMExp outputs for both ViT and BERT experiments. After initializing the decoder according to the model type, the embeddings $x^{(1)}, \cdots, x^{(K)}$ are decoded and the selected segments for exploration are extracted. These segments are then used to replace the corresponding parts of the original input instance.

---

**Algorithm 3** Interpretation for Exploration results for ViT and BERT models.

---

1: **Inputs:**
2:     Transformer model $T$ with: Patcher/Tokenizer $\mathcal{T}_T$, Embedding layer $\mathcal{E}_T$. Input image/text $z$
3:     Modified embeddings $x^{(1)} \ldots x^{(K)}$ resulted from Algorithm 1 or 2 applied on $x^{(0)} = \mathcal{E}_T(\mathcal{T}_T(z))$
4:     $P \subseteq \{1, \ldots, dim(z)\}$ indices of patches/tokens to update
5: **If** $T$ **is** ViT:
6:     Initialize decoder $d$ with weights from $\mathcal{E}_T$.
7: **If** $T$ **is** BERT:
8:     Initialize decoder with intermediate and final layers of a BERT for MLM task.
9: Decode modified embeddings $x^{(0)} \ldots x^{(K)}$ using $d$ to generate the images/sentences $Z' = Z'_0 \ldots Z'_K$.
10: **For each** $z' \in Z'$: replace segments relative to indices $P$ in $z$ with those in $z'$.
11: **Outputs:**
12: Modified input images/sentences, one for each SiMEC/SiMExp iteration.

---

Figure 1 (top left) presents the outcome of applying Algorithm 3 to a ViT exploration experiment on a CIFAR10 image. Both SiMEC and SiMExp produce visually similar outputs—each still resembling a "cat" to a human observer—yet the SiMExp interpretation is classified as "dog" at iteration 750. This demonstrates how subtle modifications, such as changes in background pixels, can significantly influence model predictions, even when such changes are perceived as irrelevant by humans, as also noted in [24]. A clear difference emerges in the exploration dynamics of the two algorithms (Figure 1, top right): SiMExp progresses in a more straight and directed manner, reflecting its goal of escaping the initial equivalence class. This divergence is further illustrated in the bottom right subplots, where the class probability distributions remain stable during SiMEC exploration but show notable fluctuations under SiMExp exploration. The lower part of Figure 1 shows a similar example for textual data from the Measuring-Hate-Speech (MHS) dataset. In this case, SiMExp identifies an alternative sentence that is classified as "Hatespeech", contrasting with the original input, which had been classified as "Offensive".

## 4 Experiments

Experiments are conducted on textual and visual data and are aimed at two objectives: *(i)* obtaining an empirical verification of the behavior of SiMEC and SiMExp under diverse settings, and *(ii)* verifying

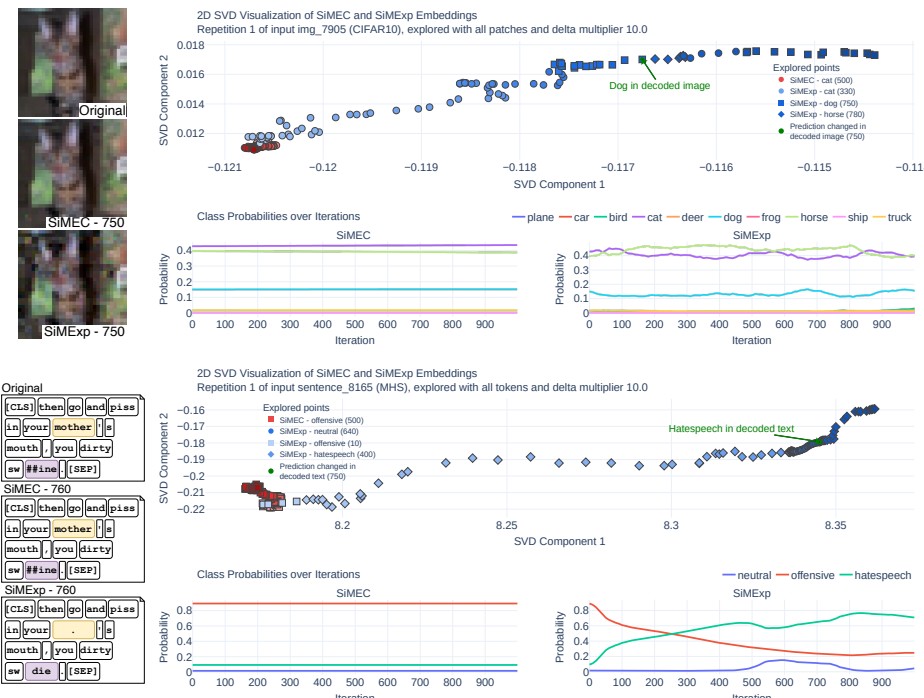

Figure 1: (**Top figure**) Example of exploration on a CIFAR10 image using SiMEC and SiMExp. *Left:* Original image, followed by interpretation outputs of $x_{750}$ from SiMEC (middle) and SiMExp (bottom). *Right top:* SVD projection of the explored points $x^{(1)}, \cdots, x^{(K)}$ for SiMEC (red) and SiMExp (blue), where color intensity encodes iteration progress (darker colors correspond to later iterations), and point shapes indicate predicted class labels. *Right bottom:* Evolution of class probabilities over iterations, for SiMEC (left) and SiMExp (right). (**Bottom figure**) Example of exploration on an MHS sentence using SiMEC and SiMExp. Visualization layout and interpretation are analogous to the top figure.

the consistency of interpretation outputs with the ones from exploration only, in order to test their usability as alternative input data.

In each data modality, we experiment with two datasets presenting different features: (*i*) MNIST [13], a grayscale digit image dataset; (*ii*) CIFAR10 [11], a RGB object image dataset; (*iii*) WinoBias [25], a textual dataset for MLM, especially focused on gender bias; (*iv*) Measuring-Hate-Speech (MHS) [16], a textual dataset for Text Classification, especially focused on hate speech detection. We trained one ViT model for each image dataset, and we used pretrained BERT models for MHS and Wino-Bias. More details about adopted models, experimental results in further configurations, and full experimental details are provided in the Supplementary Materials.

**Input space exploration** For objective *(i)* we consider the following metrics: effectiveness of the exploration can be measured by changes in prediction probabilities as well as estimation of the hyper-volume explored, while speed is assessed in terms of total time (in seconds)[2]. Algorithms are run for $K = 1000$ iterations, which we prove sufficient to capture their behavior, with delta multiplier $\eta \in \{1; 10\}$, the latter used with the aim to verify whether it is possible to speed up the process pace without compromising its stability. Finally, the experiments reported here all refer to the configuration in which all patches of an image are modified, while for textual inputs only the token with the highest attribution value is subject to exploration.

Given the predictions obtained by re-applying the models' encoder and classifier layers to the modified embeddings $x^k$ at each iteration $k$, we observe the changes in class probabilities. The theoretical

---

[2]All experiments are based on the current PyTorch implementation of the algorithms and run on a Ubuntu 22.04 machine endowed with one NVIDIA H100 GPU and CUDA 12.4.

results suggest that SiMEC should induce minimal fluctuations in them, while SiMExp should yield rapidly changing probabilities, up to prediction changes.

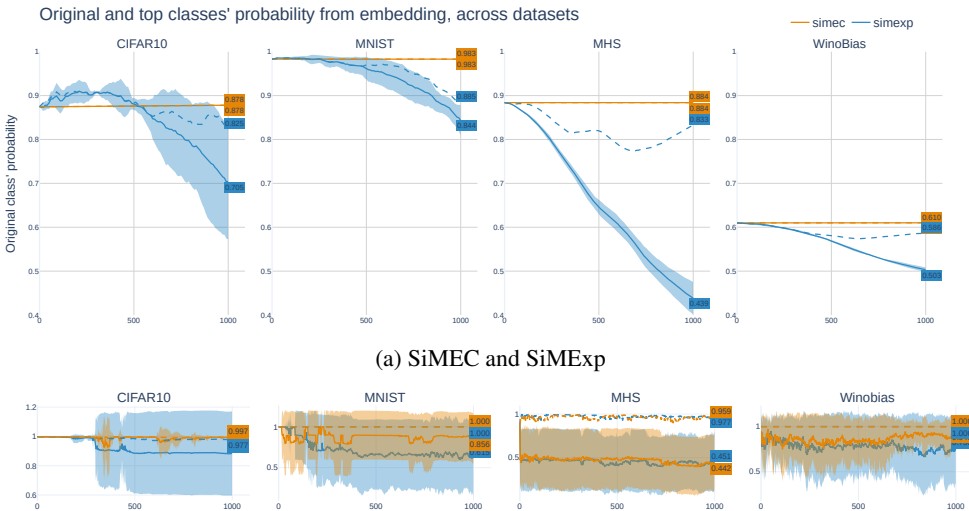

(a) SiMEC and SiMExp

(b) Baseline gradient-based Gaussian noise approaches

Figure 2: Mean and standard deviation (where applicable) of probability values for the original class (solid line) and the top predicted class (dashed line) based on embeddings obtained during exploration, across iterations and datasets. Subfigure (a) depicts the behavior of SiMEC (orange) and SiMExp (blue), while subfigure (b) reports the behavior of corresponding baseline algorithms. SiMExp results in a notable decrease in the probability of the original class, while the probability of the highest-scoring class decreases to a lesser extent, indicating a shift in the most probable class.

Figure 2a depicts the empirical reflection of the theoretical results. Focusing on the probability of the original input class (i.e. class predicted at $k = 0$), we see that SiMEC manages to keep it constant while SiMExp makes it drop significantly within the first 1000 iterations. As a baseline, we compare our results with a gradient-based Gaussian-noise approach which updates the input embedding $x^{(k)}$ at each iteration by $\pm \bar{\delta} \nabla \mathcal{N}(x^{(k)}) + \epsilon$, where the sign is determined by what exploration we are performing (same class vs other class) and $\epsilon$ is a small Gaussian noise vector orthogonal to the gradient so to guarantee exploration. Applying the same number of iterations and the same average step size $\bar{\delta}$ in each experiment allows us to conclude that SiMEC and SiMExp are significantly more effective than the baselines for staying in the original equivalence class and moving to another class respectively. Indeed, in SiMEC original class probability always follows strictly the top class probability, which in the baseline is rarely the case; in SiMExp the two probabilities tend to diverge as predicted by the theoretical analysis, while in the baseline they remain close to one another.

In order to verify the actual shift in class probabilities, we report in Table 1 the statistics about average class ranking changes, which indicate a clear tendency of SiMExp changing equivalence classes compared to SiMEC.

Furthermore, we estimate the per-patch explored hypervolume by reducing embedding vectors to the first $n$ principal components, retaining 90% of the total variance, and computing at each iteration $k$ the element-wise difference $\Delta_i^{(k)} = (\max_{t=0,\dots,k} x_i^{(t)} - \min_{t=0,\dots,k} x_i^{(t)})^{1/n}$ (the power $1/n$ allows for more stable computation). The product of the components of $\Delta^{(k)}$ gives the volume of an hyper-dimensional cuboid which contains the explored region and is thus an over-estimate of the scope achieved by the exploration. By computing the average volume ratio $\rho_V = (\Pi \Delta_{SiMExp}^{(K)} / \Pi \Delta_{SiMEC}^{(K)})^n$, we empirically verified that SiMExp explores a portion of space that is bigger than the one explored by SiMEC by an order of $10^1$. We validated these results by performing Welch t-tests on $\rho_V$: all p-values resulted lower than $10^{-3}$. Thus we conjecture that the exploration took a privileged direction on SiMExp experiments, thus making the volumes increasing faster than in SiMEC experiments.

|  |  | MNIST | CIFAR10 | WinoBias | MHS |
|---|---|---|---|---|---|
| SiMEC | $\eta = 1$ | 0.03 (0.18) | 0.0 | 0.0 | 0.0 |
|  | $\eta = 10$ | 0.07 (0.25) | 0.12 (0.33) | 0.0 | 0.0 |
| SiMExp | $\eta = 1$ | 1.31 (1.64) | 2.60 (3.35)** | 0.13 (0.33) | 0.75 (0.89)** |
|  | $\eta = 10$ | 11.91 (11.87)** | 18.70 (15.90)** | 3.25 (2.41)** | 3.64 (3.43)** |

Table 1: Average *class ranking changes* per 1000 iterations across datasets and $\eta$ values. The *class ranking change* is computed by counting pairwise inversions in class rankings before and after each exploration update. Results are reported as mean (standard deviation) over multiple runs and inputs. The symbol ** indicates that these ranking changes involve at least once a *prediction change*, i.e., a change in the top predicted class. SiMExp induces substantially more ranking and prediction changes than SiMEC, especially for larger $\eta$.

Finally, we measure the time required to explore the input space of a model with the SiMEC and SiMExp algorithms. Means (and standard deviations) of required times are computed per patch/token and per iteration: 0.126 s (0.008) for CIFAR10, 0.050 s (0.004) for MNIST, 0.300 s (0.020) for Winobias, and 0.310 s (0.074) for MHS.

**Using interpretation outputs as alternative input data**    Objective *(ii)* is to assess whether our interpretation algorithm (Algorithm 3) can generate alternative input data that either preserve the original input's equivalence class or shift to a different one, depending on the exploration dynamics of SiMEC and SiMExp. The mean difference of pixel/tokens generated from iteration to iteration amounts at $2.219 \cdot 10^{-3}$ for SiMEC experiments, and at $83.028 \cdot 10^{-3}$ for SiMExp experiments; these values increase as the number of explored patches. These results show that our algorithms generate diverse outputs across iterations, especially when the exploration is performed following the SiMExp algorithm.

Beyond output diversity, we evaluate whether and how the model's prediction for the original equivalence class evolves as SiMEC and SiMExp explore the embedding space. Specifically, we track how the class probabilities from the modified embeddings change when decoded back into the data domain every $\hat{k}$ iterations. This evaluation differs from simply observing changes in embedding predictions. The projection step (Algorithms 1 and 2, step 11) constrains modifications to an $L^\infty$ sphere containing the data domain, not the exact input space. Additionally, the decoder itself introduces approximations, either due to model limitations or numerical errors. These factors can cause discrepancies between predictions from embeddings and from decoded interpretations. To quantify this effect, we compute the average Wasserstein distance ($p = 1$) between probability distributions predicted from embeddings and their corresponding interpretation outputs. Wasserstein distance is preferred over KL divergence for its interpretability in this context. Results show that, with a median Wasserstein distance of 0.0, SiMEC produces interpretation outputs whose predicted probabilities remain very close to those of the embeddings, indicating consistent generation of reliable alternative input data. In contrast, SiMExp's outputs exhibit larger and more variable Wasserstein distances, whose distribution has a median of 0.049, highlighting inconsistencies between embeddings and their interpretations.

In cases where predictions on SiMExp's interpretations initially differ from prediction on SiMExp's embeddings, an average of 10.9% of these misaligned predictions eventually realign as exploration progresses. On average, this "catch-up" effect occurs after an average of 290.65 iterations, suggesting that longer exploration trajectories (beyond 1000 iterations) could further improve alignment. Supporting this observation, we find an average positive Pearson correlation of 0.32 (average p-value 0.08) between top class' probability predicted on SiMExp's embeddings and its corresponding probability prediction on SiMExp's interpretations, indicating a trend towards convergence.

## 5   Related work

Works dealing with embedding space exploration mostly focus on the study of specific properties of the embedding space of Transformers, especially in NLP. For instance, Cai et al. [6] challenge the idea that the embedding space is inherently anisotropic [9] discovering local isotropy, and find low-dimensional manifold structures in the embedding space of GPT and BERT. Biś et al. [4] argue that the anisotropy of the embedding space derives from embeddings shifting in common directions

during training. In the field of Computer Vision, Vilas et al. [23] map internal representations of a ViT onto the output class manifold, enabling the early identification of class-related patches and the computation of saliency maps on the input image for each layer and head. Applying Singular Value Decomposition to the Jacobian matrix of a ViT, Salman et al. [17] treat the input space as the union of two subspaces: one in which image embedding doesn't change, and another one for which it changes. Except for the last one, all the aforementioned approaches rely on data samples. By studying the inverse image of the model, instead, we can do away with data samples. The idea of applying Riemannian geometry to capture geometric information about the input manifold of a neural network building a foliation in equivalence classes has also been explored in [10, 21, 22] in the case of simple architectures. In these works a foliation of the data domain is obtained by means of the pullback of a variation of the Fisher information matrix for classifier networks with ReLU and softmax activation functions, with applications to knowledge transfer and the study of adversarial attacks.

In contrast to these works, we apply Riemannian geometry techniques to study the embedding space of transformers, computing the pullback of the metric of the output space, and we address the further problem of interpreting the output of the exploration process. Furthermore, our algorithms explore the embedding space dynamically, with a non-fixed choice of the integration step $\delta$.

## 6 Conclusions

Our exploration of the Transformer architecture through a theoretical framework grounded in Riemannian geometry led to the application of our two algorithms, SiMEC and SiMExp, for examining equivalence classes in the Transformers' input space. In particular, our method enables two complementary exploration strategies, one for retrieving input instances that produce the same class probability distribution as the original instance, the other for discovering instances that yield a different class probability distribution. We demonstrated how the results of these exploration methods can be interpreted in a human-readable form and how the exploration outputs can be used to generate alternative input data.

Future research directions include delving deeper into the potential of our framework for controlled input generation within an equivalence class. Our goal is to investigate how, in the XAI scenario, our framework can facilitate local and task-agnostic explainability methods applicable to Computer Vision (CV) and Natural Language Processing (NLP) tasks, among others. In particular, we see our methods as potential approaches to investigate Transformers' sensitivity and explainability with respect to input data features. In future applications to large-scale architectures where the dimension of the embedding space is of order $10^3 - 10^4$ [5, 19], we also plan to improve the scalability of the SiMEC and SiMExp algorithms, e.g. making use of partial decompositions.

## Acknowledgments and Disclosure of Funding

This work is partially supported by PNRR-NGEU program under MUR 118/2023.

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
