# OpenReview forum: "Unveiling Transformer Perception by Exploring Input Manifolds"
_NeurIPS.cc/2025/Conference — NeurIPS 2025 poster_

### Official Review · Reviewer_AbP5 · 2025-06-26

**Clarity:** 2
**Significance:** 3
**Originality:** 4
**Rating:** 5
**Confidence:** 3

**Summary:**

The paper tackles the question “what does a Transformer consider the same?” by searching over the model’s input space using theory from Riemannian-geometry. Specifically, each layer is treated as a smooth deformation of a manifold, and the authors pull the output metric back to the input and slice that space into equivalence classes—groups of inputs the model assigns the same class-probability distribution. On top of this theory they build two algorithms: SiMEC, which move inside a class to surface to identify new, prediction-preserving examples, and SiMExp, which tries to move away from the current class to force the model toward a different label. They also show how to decode the explored embeddings into human-readable images or sentences, and run experiments on ViT and BERT variants with CIFAR-10, MNIST, WinoBias and hate-speech datasets to check speed, volume explored, and how often predictions flip.

**Questions:**

1. It is not entirely clear to me why we need to interpret the output of the algorithms. I understand why this is needed for textual data -- we optimize over continuous embedding but want discrete tokens. But for images, presumably the space is already continuous, so why not use the output as is?

2. In text, are there examples in which multiple tokens change, or we see changes more meaningful than what is presented in Fig 1 ("Mother" --> ".")?

3. What do we learn from the WinoBias experiments? Can we shed some light on the way gender bias is manifested in the model?

**Ethical Concerns:**

["NO or VERY MINOR ethics concerns only"]

**Final Justification:**

The paper proposes a mathematically rigorous approach to understand how transformer models partition their input space. I initially had concerns about the writing and I did not understand some key points, which were clarified in the rebuttal. I hope the authors will revise the introduction according to their response.

**Limitations:**

yes

**Paper Formatting Concerns:**

None.

**Quality:**

2

**Strengths And Weaknesses:**

The paper studies an interesting question, aiming to induce minimal changes to a given input such as prediction either stays the same or flips. Beyond interpreting the input space and demonstrating how the model "sees" it -- i.e., which kind of changes correspond to output changes, and which do not -- being able to explore the model's (approximated) equivalence classes can be practically useful in debugging adversarial examples, e.g., showing that the model constructs its representation space in a non-robust way, where inputs which should be semantically equivalent end up in different equivalence classes.

The paper builds on mathematical theory, and the writing is quite dense. As someone with only rudimentary familiarity of the topic, it was quite difficult to understand the *meaning* of many of the mathematical formulations or their relevance. For instance, it was difficult to me to understand why the notion of pullback is relevant. I think I understand it now as a way to map the relation between inputs and outputs using the Jacobian, but this should be explained better. In terms of the experimental results, it is shown that the two algorithms are effective, in the sense that one of them roughly preserves output probabilities, while the others flips the class predictions. However, results are very limited, and beyond the average change in probability it is not clear whether the method is actually useful for any interpretability purpose. Do we can actually use it for debugging? What do we learn about the geometry of the input space?

One basic limitation of the proposed algorithms is that they optimize the continuous space, while we actually care about *semantic equivalence*. It's ok if the only goal is debugging -- the model might be convinced that slight pixel changes suddenly make an image of a cat look like an image of a dog -- but if the goal is to create human-interpretable adversarial examples, we would ideally be able to induce changes that bring us towards another equivalence class and go beyond slight noising of the original image.

Another issue with the evaluation is the lack of comparison with baselines. The two algorithms do achieve their goal, but one can think of other search methods that can conceivably also succeed, such as random gaussian noising and rejection sampling, gradient-guided search (without using second-order signal), etc. How do these compare with the proposed algorithms?

---

> ### Author Rebuttal · Authors · 2025-07-29
>
> We thank the reviewer for the precise remarks and questions. First of all, we want to shed more light on the general purpose of our work. The main objective of our work is to demonstrate an exploration method that adapts to the manifolds used by the transformer itself, with fewest assumptions as possible about their properties. In other words, our contribution is to theoretically formalize a new method for exploring how Transformers "see" the input space, respecting the metric of the output space. Furthermore, we want to demonstrate the empirical validity of the method using modern architecture and realistic tasks.
> The second objective concerns the analysis of the behavior of our method in practical terms, which cannot be limited to the description of the embeddings obtained from the exploration, but must be extended to include a qualitative perception of what these embeddings represent in the input space understood by the model. As a consequence, the methods presented are not specifically tailored to interpretability of models, debugging, adversarial example generation, sensitivity analysis, or bias detection, but may be used for all these downstream tasks according to purpose, which is naturally an interesting future direction. As an example, in the field of adversarial example generation, our algorithms may be used to systematically build samples that are semantically equivalent for the model, which we claim to be as crucial as building samples that are semantically equivalent for the human point of view.
>
> We now delve into the theoretical background. Other previous works exploit the Jacobian matrix of a Transformer model with the aim of exploring the input manifold. However, previous works focus exclusively on Euclidean spaces, whereas the relevance of the pullback metric relies precisely in the fact that it can be used to map the Jacobian, through equation (4) in the paper, into non-Euclidean spaces. As an example, we can think of hyperbolic-space Transformers (e.g. Ermolov et al, CVPR, 2022), in which the g matrix would correspond to a Lorentz matrix (Tabaghi \& Dokmanic, SIGKDD, 2020). In the literature, few works have treated non-Euclidean output spaces in Transformers, respecting the true geometry of the model's mapping spaces.
>
> Finally, as for the experimental section, what we learn about the geometry of the input space is fundamentally in which directions, starting from an initial data point, we can move by keeping the output constant (i.e. remaining in the same equivalence class) and which are the directions that bring to a different equivalence class in the shortest ways. While difficult to visualize in human-understandable terms, due to the high dimensionality of the input space, this can be thought of as identifying what has to be changed (and how) to shift from one class to another (see for instance the token change and concentrated patch changes in Figure 1) and how may be changed without altering how the model sees the input (in Figure 1, for example, diffused changes in the image, or a word changed with a synonym as in Figure 3 in the Supplementary Material).
> The lack of baselines is crucially due to the fact that other methods are not focused on remaining in the same equivalence class, while at most focusing on changing it assuming a Euclidean output space (gradient-guided search). While we did experiments with Gaussian noising, they revealed only that, though faster in computation, the method was useless in both objectives regarding equivalence class exploration, since it needed a huge number of iterations (at least in the order of 10000) to produce a dense enough exploration.
>
> Punctual answers to the questions are provided in the following:
>
> **Q1. It is not entirely clear to me why we need to interpret the output of the algorithms. I understand why this is needed for textual data -- we optimize over continuous embedding but want discrete tokens. But for images, presumably the space is already continuous, so why not use the output as is?**
>
> In this case we are dealing with the output of the algorithms, i.e. the embeddings found in the exploration process, and not the output of the model, i.e. the probability distribution in the output space. While we also thought about the differences between continuous and discrete decoding of embeddings at the beginning of our work, we realized that the distinction between the two cases is unnecessary, as in both cases we can think of the final transformation as a decoding step. Indeed, a decoder is necessary in all cases, with any data type processed by the model. In the textual case, data are represented as one-hot encoded vectors, as such the decoder includes a final discretizing layer. In the visual case this step is not needed, however we still need a decoder, which in the most simple case corresponds to a pseudo-inverse matrix, that maps patch embeddings back to RGB tensors. Finally, we also consider that the image space may not be intended as continuous, but rather as a highly granular discrete space, since pixels cannot really assume continuous values.
>
> **Q2. In text, are there examples in which multiple tokens change, or we see changes more meaningful than what is presented in Fig 1 ("Mother" --> ".")?**
>
> Premising that the examples, such as the one in Figure 1, report simple cases with one or two tokens modified at a time for the sake of clarity, we provide more complex experiments in the Supplementary Material, from section 5 onwards. In the tables therein presented, multiple token/patch configurations are analyzed. Specifically, for the MHS dataset: a single token, all tokens simultaneously, as well as half of them. The example provided in Figure 1 was most significant for SiMExp, while the reviewer may find an interesting example for SiMEC in Figure 3 of the Supplementary.
>
> **Q3. What do we learn from the WinoBias experiments? Can we shed some light on the way gender bias is manifested in the model?**
>
> As previously stated, bias detection may be an interesting downstream task of our proposed approach. In here, WinoBias dataset is used as a practical, real-world example of the empirical validity of the SiMEC and SiMExp algorithms, not much for studying the problem of gender bias. In other words, while gender bias detection is certainly an interesting topic in terms of applicability of our approach, which we deem promising for the specific task, it would require further and more focused research.

---

> > ### Comment · Reviewer_AbP5 · 2025-08-03
> > **Response**
> >
> > Thank you for the detailed response. It answers my concerns. I urge the authors to restructure parts of the paper, particularly the introduction and the description of the method, to make the goal and the conclusions drawn from the developed methods clearer. I raised my score to "accept".

---

> > > ### Author Response · Authors · 2025-08-04
> > > **Thanks to the reviewer**
> > >
> > > We sincerely thank the reviewer for their time, availability, and valuable feedback. Their thoughtful comments have been instrumental in helping us improve the quality and clarity of our work. We are committed to incorporating these insights into the final version of the paper and believe they will significantly strengthen both its rigor and readability.

---

### Official Review · Reviewer_JwKY · 2025-07-03

**Clarity:** 2
**Significance:** 3
**Originality:** 3
**Rating:** 4
**Confidence:** 3

**Summary:**

The paper proposes a new interpretability tool for Transformers through the lens of Riemannian geometry. By modeling Transformer as smooth maps between manifolds, the authors compute the pullback of the output space metric onto the input space via the model’s Jacobian. This enables two complementary procedures: (1) identifying equivalence classes of inputs that yield the same output distribution, and (2) discovering directions that lead away from the current equivalence class. These resulting methods allow for principled exploration of the embedding space, enabling controlled and interpretable input perturbations.

**Questions:**

Those are written in the weakness section above.

**Ethical Concerns:**

["NO or VERY MINOR ethics concerns only"]

**Final Justification:**

I have read the rebuttal and the answers to my questions. I appreciate that the authors took the time to clarify my questions. I still believe it is above acceptance threshold, I keep my rating.

**Limitations:**

Limitations are addressed throughout the paper.

**Quality:**

3

**Strengths And Weaknesses:**

Strengths
The use of differential geometric tools for Transformer analysis is a promising and underexplored direction in interpretability. This work contributes valuable theoretical insights to that space.
The proposed framework is modality-agnostic, demonstrated on both textual (BERT) and visual (ViT) architectures. Extending to other modalities such as audio or multimodal models would be a compelling future direction.
The projection of explored embeddings back into human-interpretable space (via decoding or inverse mapping) allows meaningful qualitative insight.
The mathematical formulation is rigorous and well presented, with clear derivations and thoughtful use of geometric formalism to support the validity of the method.

Weaknesses
-The methods works entirely in the embedding space and the projection back to input space is approximated and lossy. The methods practical impact e.g. for adversarial examples is therefore limited by the quality of the inverse mapping
-The paper lacks comparisons to alternative interpretability methods. While the approach is conceptually distinct, a baseline or ablation comparison would better situate its effectiveness.
-The authors state that Transformer components, such as fully connected layers, GeLU, and sigmoid activations, are smooth functions. Does that also hold true for other essential components, such as the softmax operation?
-The method is evaluated only on one architecture per modality, and only on small or toy datasets. To support the claim of a general framework, broader empirical validation across architectures and tasks would be beneficial .

Minor weakness:
Some sentences are overly dense or overloaded with technical detail, affecting clarity (e.g. 140-143).  A clearer writing style would make the paper more accessible to a broader audience.

---

> ### Author Rebuttal · Authors · 2025-07-29
>
> We thank the reviewer for the punctual remarks. As the precise questions are reported in the "weaknesses" section, we try to summarize and provide exhaustive answers to the points highlighted in there.
>
> **Q1. It is highlighted that projection from the embedding space to the input space is lossy**
>
> We acknowledge this point in the paper, however we stress that one key point of our experiments is precisely to evaluate the difference produced by mapping the embedding vectors back into the input space through the interpretation step. Specifically, producing adversarial examples was not our primary objective, although it is an interesting application of our algorithms, just like sensitivity analysis and model explainability. From the empirical analysis we conducted in the paper, however, it is possible to support the claim that the theoretical framework can be applied effectively to real data, as well as to draw meaningful qualitative insights, such as visualization of changes in the input space (image patches or textual tokens in the experiments) by moving along relevant directions (i.e. those of maximum and null output change) in the embedding space.
> Furthermore, we believe human interpretability of the output of the algorithms to be an important feature as well, which can be achieved only through decoding, as it would be unfeasible to compute the pullback metric directly on high-dimensional input vectors such as vocabulary size for textual tokens.
>
> **Q2. Ablation study**
>
> As for ablation study, since all steps of our algorithms are fundamental in its inner functioning, we refer the reviewer to tables from Section 6 onwards of the Supplementary Material, which provide a comparison between different choices of delta multiplier ($\eta = 1$, which corresponds to eliminating the multiplier, and $\eta > 1$, which aims at accelerating the exploration process, while we also experimented with $\eta < 1$, without relevant results) and fraction of patches/tokens explored (a single patch/token, all patches/tokens simultaneously, half of them).
>
> **Q3. Is softmax a smooth function?**
>
> Finally, we remark that all components of most widespread Transformer Language and Vision Models are smooth functions (except for null-measure sets), including softmax and similar activations.

---

> > ### Comment · Reviewer_JwKY · 2025-08-05
> >
> > I appreciate the authors responses and clarifications. When I referred to comparison to baselines I meant to other methods that even though do not work at the embedding space or have a different methodology, have a similar objective.
> > Even though, as preliminary stated in the initial review a more general framework to support the validity of the experiments is missing and that the method is evaluated only on one architecture per modality, and only on small or toy datasets I still believe the paper is above threshold and worth publishing since it might spark future work. For this reason I maintain my score.

---

> > > ### Author Response · Authors · 2025-08-06
> > > **Thanks for supporting our work**
> > >
> > > We thank the reviewer for their thoughtful follow-up and for their support of our work's publication. We sincerely appreciate their perspective on its value in potentially sparking future research.
> > >
> > > The reviewer's clarification on baselines is particularly helpful, and we agree that comparing our work to methods with similar objectives strengthens the contribution. While our paper already provides a thorough ablation study on our method's parameters and search space, we recognize the value of an external baseline. For this reason, we will integrate a planned experiment comparing our approach against a Gaussian noise-based random walk. This is a deliberate and essential first step, designed to rigorously validate the core necessity of our geometry-aware framework before expanding to comparisons against methods with fundamentally different methodologies.
> > >
> > > This same principle of establishing a clear foundation guided our overall experimental design. We agree with the reviewer's assessment that our work is foundational, and our use of vision and text Transformer models on well-known datasets was a strategic decision made precisely to this end. This approach allows us to provide an unambiguous validation of our method's core principles and range of operability. We believe this principled and focused validation establishes the necessary grounding for the future applications that we and the reviewer envision.
> > >
> > > We thank the reviewer once again for their constructive feedback and support.

---

### Official Review · Reviewer_KszG · 2025-07-05

**Clarity:** 2
**Significance:** 3
**Originality:** 2
**Rating:** 4
**Confidence:** 3

**Summary:**

The authors present an information geometric method for exploring how Transformer models encode their input data by identifying *equivalence classes*. i.e sets of inputs that yield the same or different predictive outcomes. In particular, the paper relies on Riemannian geometry to treat the layers of a Transformer as a sequence of transformations of the input manifold. The primary contributions are two algorithms:
1. Singular Metric Equivalence Class (SiMEC): Identifying inputs that belong to the same equivalence class.
2. Singular Metric Exploration (SiMExp): Complementary to SIMEC, finding perturbations to input that can change the model's prediction is important to understand the robustness of features. SIMEXP identifies inputs that to change equivalence classes, yielding a different probability distribution.

At a high level, the core insight is based on the spectral decomposition of a "pullback metric" that translates the geometry of the model's output space back to its input space. By moving along eigenvectors associated with zero eigenvalues (i.e. nullspace for SiMEC) or non-zero eigenvalues (for SiMExp), the algorithms can systematically explore the model's input manifold. The authors also demonstrates how these perturbed embeddings can be decoded back into human-readable formats like images and text, providing interpretable insights into the model's decision boundaries.

**Questions:**

Some questions that would help improving understanding the core contributions of the paper

1. Computational overhead : Per my understanding, the eigendecomposition of the pullback metric has a computational complexity (per-token) of O(n^3). For modern architectures like ViTs, the models have embedding dimension in the order of 1024+. Could you please elaborate on potential ways to scale the method (e.g. via approximate eigendecompositions)?

2. Impact of the feasible region projection: Could you share a more details on the design choices of the projection, such as the sensitivity of this step? E.g. how do the results change if a different projection method is used, such as projecting to the nearest neighbor in the training set's embedding space or onto a learned manifold (e.g., via a VAE)?

3. generalizing the textual exploration beyond a single token : The experiments on the text modality provide a compelling proof-of-concept, but the scope is limited to modifying only the single token with the highest attribution value. Could the geometric information from the eigenvectors from this framework be leveraged to perform a more holistic exploration of textual inputs involving changes to multiple tokens?

**Ethical Concerns:**

["NO or VERY MINOR ethics concerns only"]

**Final Justification:**

The paper's core contribution is its novel information-geometric framework for analyzing Transformer manifolds, which is technically sound. The rebuttal provides confidence that the main implementation limitations (scalability, fidelity) are addressable with future engineering, not inherent theoretical defects. I maintain my score at 4 (Borderline accept), reasons to accept outweigh reasons to reject.

**Limitations:**

yes

**Paper Formatting Concerns:**

no major formatting issues

**Quality:**

2

**Strengths And Weaknesses:**

### **Strengths**
Some strengths of the paper as submitted

1.  **Principled and well-defined problem statement**: The paper's methodology builds on a robust and formally-defined foundation of Riemannian geometry. By treating a neural network as a sequence of maps between manifolds and using the pullback of the output space metric to define a singular Riemannian metric on the input space, the authors establish a formal framework to define equivalence classes based on zero pseudodistance ($Pd(x,y)=0$).

2.  **Geometry based exploration across equivalence classes**: The **SiMEC** and **SiMExp** algorithms are a direct and innovative application of the paper's geometric theory. The insight of using the eigendecomposition of the pullback metric ($g^0_x$) and separating the exploration into directions within the kernel (eigenvectors of the zero eigenvalue) versus its complement (eigenvectors of non-zero eigenvalues) is useful for navigating within or across equivalence classes, respectively.

3.  **Interpretability via input manifold exploration**: A significant contribution is the concrete method (Algorithm 3) for decoding explored embeddings back into a human-understandable format for both vision and text. The auhors provide practical approaches to recover such readouts based on certain constraints and assumptions.

4.  **Empirical validation**: The paper's claims are validated across different data modalities (vision and text) and datasets (MNIST, CIFAR-10, WinoBias, and MHS). The experiments are well-designed to test the core hypotheses.


### **Weaknesses**

1.  **Computational overhead**: The primary limitation of the proposed method is its computational complexity. The reliance on eigendecomposition of the pullback metric has a complexity of $O(n^3)$, where *n* is the embedding dimension. This makes the approach computationally expensive and potentially impractical for state-of-the-art Transformers, which often have large embedding dimensions (e.g., 768, 1024, or higher).

2.  **Potentially strong assumptions**: The practical implementation as described above involves several critical approximations that is likely to affect fidelity. The step size $\delta^{(k)}$ is an estimate based on a local linearity assumption, and the projection onto a "feasible region" is a bounding box acknowledged as an over-approximation. These numerical errors can accumulate, leading to inconsistencies between the predictions from embeddings and the corresponding decodings might be misleading.

3.  **Limitations to single-token exploration**: The text-based experiments are constrained to exploring modifications of only a **single token** with the highest attribution value. This is a significant simplification, as true semantic equivalence or difference in natural language often involves complex, combinatorial, or syntactic changes across an entire sentence.

---

> ### Author Rebuttal · Authors · 2025-07-29
>
> We thank the reviewer for their insightful feedbacks on our work.  We appreciate that the reviewer acknowledges both our theoretical contributions and our empirical validation of the method we develope to explore how transformers encode their input data. The reviewer's remarks raised some interesting questions which we address below.
>
> **Q1. Computational overhead : Per my understanding, the eigendecomposition of the pullback metric has a computational complexity (per-token) of** $O(n^3)$ **. For modern architectures like ViTs, the models have embedding dimension in the order of** $1024+$ **. Could you please elaborate on potential ways to scale the method (e.g. via approximate eigendecompositions)?**
>
>  In our work we decided to use the full decomposition of the metric to study our method in its entirety, thus considering all the information about the embedding space geometry carried by the spectral decomposition of the pullback metric. However, our algorithms can be modified to scale better with the embedding dimension. One possibility -- namely using an approximate eigendecomposition -- is suggested by the reviewer. While this method is indeed an option to make the problem scale better, it also introduces additional numerical errors at each step of the exploration process, whose accumulation may be not negligible in the long run. To avoid this problem, our method can be scaled making use of partial decompositions (such as Lacnzos algorithm and its variants like spectrum slicing techniques) computing e.g. only the first $k$ highest eigenvalues and associated eigenvectors (for SiMExp) or the lowest $k$ ones (for SiMEC). These methods, albeit potentially restricting the exploration of the embedding space to a smaller volume compared to the full decomposition of the metric, may also speed up the exploration process of in SiMEXP (the directions corresponding to the highest eigenvalues correspond to more modifications to the input). For SiMEC, limiting the eigendecomposition to the lowest eigenvalues can decrease the approximations error in the path built by SiMEC (the directions corresponding to the lowest eigenvalues correspond to less modifications of the input equivalence class) possibly at the expense of the speed of exploration.
>
> **Q2. Impact of the feasible region projection: Could you share a more details on the design choices of the projection, such as the sensitivity of this step? E.g. how do the results change if a different projection method is used, such as projecting to the nearest neighbor in the training set's embedding space or onto a learned manifold (e.g., via a VAE)?**
>
> We agree with the reviewer's comment about our feasible region projection method that the projection to a bounding box is an over-approximation. However, it is a computationally inexpensive method which does not affect the speed of exploration and does not restrict the explored volume. Projecting to the nearest neighbor in the training set at each step reduces the speed and the volume of the exploration, since only significant enough changes of the input may be projected to different token embeddings. Small values of delta or proceeding in directions corresponding to small changes of the starting embedding may lead to a situation in which the projection leads back to the starting point, effectively preventing the exploration of the embedding space. Making use of a VAE to project to the feasible region is indeed an interesting suggestion which has been proposed for other (non-transformers) architectures employed to solve inverse problems in medical imaging. While worth experimenting with, in our work we decided to focus on the standard transformer architecture, hence the decision not to use additional networks to perform the projection.
>
> In their comments, the reviewer also addressed another point related to this question, namely the validity of the hypotheses under which we perform the estimate of the step size $\delta^{(k)}$. We think that the local linearity assumption is a reasonable hypothesis due to the fact that smooth manifolds are locally Euclidean and our metric is smooth (and therefore can be locally approximated by a Taylor expansion within a bounded error). While we agree with the reviewer that the estimate of the step size may introduce numerical errors, we note that this also happens e.g. in Euclidean ODEs integration algorithms such as the standard Euler method -- assuming that the solution can locally be approximated by a polygonal curve, which is indeed a local linearity assumption. We are aware that the accumulation of the numerical errors may lead to inconsistencies between the predictions from the embeddings and the corresponding decodings. We addressed this problem in Section 3.2 of our paper and in Section 6 of the Supplementary Materials, carrying out some numerical experiments to study the effect of decoding the embeddings produced by the algorithms.
>
>
>
> **Q3. Generalizing the textual exploration beyond a single token : The experiments on the text modality provide a compelling proof-of-concept, but the scope is limited to modifying only the single token with the highest attribution value. Could the geometric information from the eigenvectors from this framework be leveraged to perform a more holistic exploration of textual inputs involving changes to multiple tokens?**
>
> Our algorithms can indeed modify more than one token, as the eigenvector decomposition carries information on the whole sequence of tokens given in inputs. In our examples we decided to limit the exploration to one or two tokens for the sake of clarity, e.g. in Section 4, Figure 1 where only two tokens are modified. However, we considered modifications to all the tokens of a sentence in the analysis carried out in Section 6 of the Supplementary Materials.

---

### Official Review · Reviewer_bq1s · 2025-07-06

**Clarity:** 4
**Significance:** 2
**Originality:** 1
**Rating:** 4
**Confidence:** 3

**Summary:**

This work develops tools using differential geometry to understand how changes to the input manifold translates to changes in the output. manifold. In particular, the authors interpret a neural network as a sequence of transformations and divide the inputs into equivalence classes depending on if they produce the same output. The authors present two algorithms to move across equivalence classes or stay within the same equivalence class, which they use to analyze networks trained on different text and vision tasks.

**Questions:**

1. I found the SimEXP algorithm to be remarkably similar to signed-gradient techniques for adversarial samples. Can the authors comment on the similarities or differences to literature on adversarial samples? Why is it surprising in this context that SimEXP works?
2. In similar spirity, one would expect small changes to the input to not change the outputs. Isn't it unsurprising that SimEC is able to find very similar looking images that make the same predictions?
3. What is the distance traveled in weight space for SimEC and SimExp. Is the distance traveled by SimEC extremely small, in which case aren't the results to be expected?

**Ethical Concerns:**

["NO or VERY MINOR ethics concerns only"]

**Final Justification:**

I've increased my score after the rebuttal since some of the concerns regarding the framework were addressed. However, I think the paper lacks evidence showing that their theoretical framework and algorithms (SimEC, SimExp) can used to provide *new* insights about neural networks or the input data used to train them.

I lean (only slightly) towards acceptance because the paper has interesting elements and there are exciting directions for future work. At the same time, I'm not extremely excited about the paper since the utility of their framework is mostly left to future work.

**Limitations:**

The main limitations are addressed in the paper.

**Paper Formatting Concerns:**

No formatting concerns

**Quality:**

2

**Strengths And Weaknesses:**

**Strengths.**
The authors do a great job of presenting the content, and provide an interpretation of neural networks as a series of deformable transformations. While the usage of the Jacobian is not necessarily new (for example literature on adversarial samples or work on interpretability like GradCAM), casting these ideas through the lens of differential geometry was interesting. I found it interesting that Sim-Exp explores a far larger volumes compared to SiMEC.

**Weaknesses**.
While I found the mathematical tools that were developed to be sound, I found the experiments section were lacking many results and found it hard to justify the utility of the developed tools. The results on SimExp seem well understood from the lens of adversarial samples. There exist adversarial samples found around an epsilon-ball at every point so it isn't surprising that Sim-Exp can find such a sample. I also find results with SimEC somewhat underwhelming since one would expect that moving along the null-space of the Fisher would result in no changes to the outputs. With this context, I found objective (i) to be unsurprising.

I found objective (ii) to be confusing. The results don't really suggest using the outputs of SimEXP or SimEC as alternative input data, but only that we can generate inputs / outputs from the same or different class. This does not suggest that we can use these inputs or outputs as additional training data. Could the authors clarify what they mean by objective (ii) and how the experiments are in support of this objective.

In general, I would have liked to seen an analysis comparing different architectures or more specific insights about the shape of the input manifolds and the equivalence classes. In my opinion, in its current state, the experiments don't really do justice to the tools developed in this work.

---

> ### Author Rebuttal · Authors · 2025-07-29
>
> We thank the reviewer for their effort and thoughtful feedback on our work. We appreciate that the reviewer recognizes the novelty of casting our analysis through the lens of differential geometry. We would like to address their concerns and clarify the contributions of our experiments.
>
> The central contribution of our paper, as the reviewer notes, is the employment of differential geometry. This allows our exploration algorithms to adapt directly to the learned manifold of the Transformer. Unlike methods that compute gradient-based updates in a presumed Euclidean space, our approach respects the specific, non-Euclidean geometry created by the model's series of transformations. This is achieved through the pullback metric, which precisely maps movement on the output manifold to the corresponding movement on the input manifold. This geometric awareness is also why a single exploration step in SiMExp can traverse a significantly larger volume than in SiMEC, even with an identical step multiplier ( $\eta$ ), as it follows the natural curvature of the representation space.
> We will now address the reviewer's specific questions.
>
> **Q1. Can the authors comment on the similarities or differences to literature on adversarial samples? Why is it surprising in this context that SiMExp works?**
>
> While both SiMExp and signed-gradient techniques leverage Jacobians, the substantial difference lies in our computation of the pullback metric. This component is fundamental, as it allows us to map an exploration step from the output manifold back to the input manifold, reflecting the true geometry of the model's learned space, even if not Euclidean.
>
> Regarding why the success of SiMExp is noteworthy: while we provide a solid mathematical foundation, empirical validation is essential. It is not a given that these theoretical tools will perform as expected on complex architectures like Transformers, especially considering numerical approximation challenges (e.g. float64 or float32). To demonstrate the practical utility of our modality-agnostic method, which is designed to work across Transformer architectures, we intentionally evaluated it on two distinct models, one for text and one for images. This cross-modal evaluation highlights the robustness and generality of our approach. Empirical evidence that our method performs well in practice is a necessary complement to its theoretical foundation. These results also provide insight into the model behaviors that our geometric tools are capable of revealing.
>
> Our primary objective in this work is to validate, both theoretically and empirically, the foundations of our geometric exploration framework applied to the Transformer architecture. Given this focus, we do not position our work as a direct competitor to methods aimed specifically at adversarial example generation or data augmentation. Instead, our methodology serves a different purpose: establishing a principled understanding of model behavior grounded in geometry. We see adversarial generation and augmentation as promising downstream applications of our approach, but not the focus of this initial study. In our view, building a solid and interpretable foundation is a necessary first step to enable stronger, more reliable applications in the future.
>
> This leads directly to Objective (ii). The reviewer asked for clarification on this objective and its connection to the experiments. SiMEC and SiMExp are designed to generate new data points directly in the input embedding space, which can be used as alternative inputs in embedding form. However, this does not translate to generating corresponding samples such as new images or new texts.
> A natural next step is to project these alternative embeddings back into a human-interpretable space. This serves two key purposes. The first is qualitative interpretation: it allows us to visually inspect the generated data and observe how the exploration process alters the original input. This provides insight into what the model considers to belong to the "same" or "different" equivalence class, offering a glimpse into the structure of these classes (see token and patch changes in Figure 1). The second purpose is validation. This step is crucial to evaluate whether the human-interpretable counterparts can be used interchangeably with the embeddings from which they were generated, which involves analyzing the model's output when fed the decoded sample to understand its relationship with the output from the embedding itself.
>
> **Q2. Isn't it unsurprising that SiMEC is able to find very similar looking images that make the same predictions?**
>
> As shown in literature (e.g., Shi et al., 2022, "Decision-based black-box attack against vision transformers via patch-wise adversarial removal", NIPS '22), even imperceptible changes can drastically alter a model's prediction if made in a specific, adversarial direction.
> In this context, the contribution of SiMEC is its ability to perform a controlled exploration that selectively finds the input changes that maintain the output probability distribution. It successfully isolates directions of invariance from directions of high sensitivity. The fact that SiMEC automatically discovers the appropriate (and smaller) step size needed to stay within the same equivalence class—while SiMExp automatically finds a larger step to change it—is further empirical evidence validating our theoretical framework.
>
> **Q3. What is the distance traveled in weight space for SiMEC and SiMExp? Is the distance traveled by SiMEC extremely small, in which case aren't the results to be expected?**
>
> This is an excellent question that points to a potential misunderstanding. Our methods, SiMEC and SiMExp, operate in the input embedding space, not the model's weight space. The model's weights remain frozen during our analysis. We apologize if this was not clear and will explicitly state it in the revised methods section.
>
> The reviewer's question is likely about the distance traveled in the input space. As reported in lines 267-277, SiMExp explores a volume on average 10x larger than SiMEC. While the distance for SiMEC is smaller, the crucial point is the direction of travel, not just the magnitude. Adversarial literature shows that even infinitesimal steps can change the output if the direction is orthogonal to the decision boundary. SiMEC's contribution is finding the specific directions (tangent to the equivalence class manifold) where larger steps can be taken without changing the output. The fact that our geometrically-derived algorithms automatically discover these distinct scales of movement (small for SiMEC, large for SiMExp) further validates our framework.
>
> We thank the reviewer again for their insightful feedback. We are confident that the revisions guided by their comments will significantly strengthen our paper's contribution by better framing it as a foundational work. Our goal is to validate a new theoretical framework for manifold exploration, which we see as a necessary first step for developing powerful future applications in areas like adversarial analysis and training augmentation.

---

> ### Comment · Reviewer_bq1s · 2025-08-02
> **Thank you for the response!**
>
> Thank you for the detailed response to all the questions! I have a few more follow up questions
>
> > While both SiMExp and signed-gradient techniques leverage Jacobians, the substantial difference lies in our computation of the pullback metric.
>
> Thank you for this clarification. Signed gradient doesn't make use of the curvature of the function while the pullback metric does (with the latter being more expensive to compute). The response was helpful in clarifying this misconception on my part.
>
> > SiMEC and SiMExp are designed to generate new data points directly in the input embedding space, which can be used as alternative inputs in embedding form.
>
> It is still not clear to me what the alternative inputs (line 282) can be used for. I think there is clear evidence that SiMEC and SiMExp are able to stay within or move across equivalence classes. But I don't see any experiment in the paper that shows that we can extract insights about the model or we can use this alternative inputs meaningfully.
>
> > It successfully isolates directions of invariance from directions of high sensitivity.
>
> I think its clear that SimEC does this. But I worry that this is not difficult since most directions have low sensitivity (spectrum of Hessian has very sharp decay, i.e., most eigenvalues are small, and I expect spectrum of pullback metric to be no different). If we move a very small magnitude in a random direction, I expect the output to not change. In this light, SimExp carries out a more impressive task of moving across manifolds.
>
> I think having some baselines to contrast with SimEC would be extremely helpful. Why is SimEC necessary if a small random perturbation (in input space) could potentially achieve the same effect? It would be nice to show that SimEC can move large distances  in input space while maintaining invariance --- this is something a trivial algorithm can't achieve.
>
> > The reviewer's question is likely about the distance traveled in the input space
>
> Yes, the questions was indeed intended for the input space. Thanks for reinterpreting the question and clarifying a potential misunderstanding.
>
>
> A few of my questions were answered so I'd be happy to bump up my score. I'll wait for final response before doing so. While the theoretical framework is very interesting, I think a couple convincing experiments showing a clear path to how SimEC and SimEXP can be used is missing. Right now, most experiments are geared towards validating the proposed framework.

---

> > ### Author Response · Authors · 2025-08-04
> > **Addressing a few remaining observations**
> >
> > We thank the reviewer for their thoughtful feedback, which has helped improve the clarity and impact of our work. Below, we address the remaining concerns.
> >
> > > I think having some baselines to contrast with SiMEC would be extremely helpful. Why is SiMEC necessary if a small random perturbation (in input space) could potentially achieve the same effect? It would be nice to show that SiMEC can move large distances in input space while maintaining invariance --- this is something a trivial algorithm can't achieve.
> >
> > We agree that a comparative analysis against a baseline, such as an iterative random perturbation, is valuable. The critical distinction between SiMEC and such a baseline lies in SiMEC's explicit awareness of the underlying data geometry. The decomposition of the pullback metric in SiMEC is aware of the non-Euclidean geometry of the data, which is absent when considering random Gaussian perturbations. This leads to two primary advantages.
> >
> > First, an iterative random perturbation in input space offers no guarantee of preserving the equivalence class over time—small changes can accumulate and shift the input out of class. In contrast, SiMEC ensures movement stays within the original equivalence class by construction, unlike Euclidean noise, which may drift in directions that break class invariance.
> >
> > Second, the standard Euclidean random noise and the non-Euclidean random walk of SiMEC differ in exploration speed and the average radius of the explored volume (proportional to $\sqrt{k}$ for the Euclidean case, and a function of the metric and $k$ for the non-Euclidean scenario). In an $n$-dimensional Euclidean space, exploration has no preferred directions. Since equivalence classes are submanifolds of dimension less than $n$, a Gaussian random walk has a low probability of staying on the class manifold; its exploration is not concentrated there. Consequently, the explored volume inside the class is limited compared to a geometry-aware approach. The exploration is less dense because the walk may leave the equivalence class for a time before returning. In a non-Euclidean random walk, the Riemannian metric allows for preferred directions—in SiMEC, these are precisely the ones that preserve the class. The outcome is that SiMEC explores larger volumes inside an equivalence class because it never leaves it.
> > To empirically validate these advantages, we plan to augment the manuscript with experiments showing:
> > - A direct comparison of the distance traversed by SiMEC versus a Gaussian random walk before the latter fails to maintain equivalence class membership.
> > - On a synthetic dataset with a known manifold, a demonstration of SiMEC's adherence versus the baseline's deviation.
> >
> > > But I don't see any experiment in the paper that shows that we can extract insights about the model ...
> >
> > We will revise the manuscript to clarify that model insights arise directly from the exploration process. As shown in Figure 1 (main text) and Figures 1 and 2 (supplementary), tracking class probabilities during SiMEC’s exploration reveals which classes the model sees as semantically related, offering a local view of its decision boundaries and internal representations.
> >
> > > ... or we can use this alternative inputs meaningfully.
> >
> > Finally, regarding case studies, we agree our framework holds potential for applications like data augmentation and adversarial example generation. We chose not to include them to maintain the manuscript’s focus and clarity for two main reasons.
> >
> > First, to avoid ambiguity regarding our core contribution. Our central aim is to introduce and validate our framework as a sound method for exploring equivalence classes. Introducing downstream tasks could risk distracting the reader, causing them to conflate our method's validation with a proposal for a new state-of-the-art technique in an applied domain.
> >
> > Second, our framework is relevant to diverse and numerous applications. A meaningful analysis of any single case study would be limiting, as it could only showcase a fraction of the method's wide-ranging utility. A superficial discussion of a few applications would not do justice to their complexities, while a thorough treatment of all is beyond the scope of a single manuscript.
> >
> > Therefore, our empirical validation was deliberately limited to demonstrating that the explored embeddings behave consistently with our theoretical framework, which in turn implies their utility as alternative inputs. We will, however, incorporate a more thorough discussion in the final version of our manuscript that highlights these promising applications. Indeed, a rigorous investigation of these case studies constitutes a primary direction for our future work and, given the required depth, warrants its own dedicated study. We trust this approach will strengthen the paper by framing our current contribution as the foundational methodology upon which this subsequent applied research will be built, while maintaining its intended focus.

---

### Note · Authors · 2025-08-12

We thank all reviewers whose feedback and dialogue will help us present our contribution more effectively and improve the final manuscript.

Thanks to the reviewers’ comments, we will be able to refine the way we explain our work’s objectives: introducing a differential geometry framework for exploring Transformer models. Such models are known not to learn a space that can be adequately described by Euclidean geometry. We propose using the pullback metric to navigate the input manifold while respecting this geometry, providing formally guaranteed exploration of equivalence classes, something that approaches based on Euclidean geometry cannot achieve.

Although our methods offer formal guarantees and are architecture-agnostic, we deemed it crucial to demonstrate their empirical effectiveness on real Transformers. First, we needed to validate that implementation challenges don't undermine our techniques' effectiveness. Second, we wanted to explore whether both the embedding-space results and their decoded, human-interpretable counterparts can be meaningfully used, by analyzing how the model's outputs relate across both representations. Our experiments across different modalities provide robust validation of these aspects.

We fully agree with the reviewers' suggestions for downstream applications (e.g. data augmentation, adversarial analysis, and bias detection). Although we considered our method applications from the outset, we recognized this vast field warrants dedicated study. We therefore omitted this investigation to keep focus on our main contribution and avoid superficial treatment due to space and coherence constraints.

Nonetheless, the dialogue with the reviewers has provided us with excellent ideas that we will apply in the final version of our work, by extending the experimental section to incorporate a new baseline comparing our geometry-aware exploration method (SiMEC) with a standard Gaussian random walk. This experiment is designed to empirically demonstrate that SiMEC can traverse significantly greater distances within an equivalence class thus underscoring the necessity of our geometric approach, also in view of its potential applications.
We are confident that these revisions will address the reviewers' remaining concerns, strengthen our work, and establish the necessary foundations for future applications envisioned both by us and by the reviewers.

---

### Decision · Program_Chairs · 2025-09-17

**Decision:**

Accept (poster)

**Comment:**

This paper proposes an approach for exploring the input space of Transformers using tools from differential geometry. The main idea is to model the network's layers as a series of deformations on the input manifold. By computing the pullback of the output space metric via the model's Jacobian, the authors define equivalence classes. They then propose the algorithms of SiMEC and SiMExp, which use the eigendecomposition of this pullback metric to navigate these equivalence classes, hence providing a principled way to interpret the model's perception.

In terms of strengths, reviewers noted the paper's novel and principled theoretical framework. The aproach was highlighted as a promising, sound, and innovative  (bq1s, KszG, JwKY, AbP5). The mathematical formulation was considered rigorous and well-presented (bq1s, JwKY). The framework was noted for its practical advantages, including being modality-agnostic and providing meaningful, interpretable outputs by decoding explored embeddings (KszG, JwKY).

Inititially, a shared concerns was the lack of demonstrated utility and missing baslines. Reviewers argued the experiments were limited to validating the method's mechanics and did not show what new insights could be learned about the model or how the generated inputs could be used meaningfully for downstream tasks (bq1s, AbP5). The absence of a simple baseline, such as a random Gaussian walk,  was also noted (bq1s, AbP5). Other concerns included the computational overhead limited scope of the experiments (KszG, JwKY).

The post-rebuttal discussion solidified a clear but conflicted consensus among the reviewers. On one hand, reviewers acknowledged that the paper presents a "technically sound" and "novel information-geometric framework" (KszG) with "interesting elements and... exciting directions for future work" (bq1s). On the other hand, they were unenthusiastic, noting that the "utility of their framework is mostly left to future work" and that it "lacks evidence showing that [it] can used to provide new insights" (bq1s). The AC believes that while the immediate utility is not shown, the high-quality foundational methodology is novel, sound, and promising. Therefore, the AC finds that reasons to accept outweigh reasons to reject and recommends acceptance.